# Spectral-Based Fault Detection Method in Marine Diesel Engine Operation

**DOI:** 10.3390/s25185669

**Published:** 2025-09-11

**Authors:** Joško Radić, Matko Šarić, Ante Rubić

**Affiliations:** Faculty of Electrical Engineering, Mechanical Engineering and Naval Architecture in Split, University of Split, 21000 Split, Croatia; radic@fesb.hr (J.R.); ante.rubic@fesb.hr (A.R.)

**Keywords:** accelerometer, acoustical microphone, diesel engine, fault detection

## Abstract

The possibility of developing autonomous vessels has recently become increasingly interesting. As most vessels are powered by diesel engines, the idea of developing a method to detect engine malfunctions by analyzing signals from microphones placed near the engine and accelerometers mounted on the engine housing is intriguing. This paper presents a method for detecting engine malfunctions by analyzing signals obtained from the output of a microphone and accelerometer. The algorithm is based on signal analysis in the frequency domain using discrete Fourier transform (DFT), and the same procedure is applied to both acoustic and vibration data. The proposed method was tested on a six-cylinder marine diesel engine where a fault was emulated by deactivating one cylinder. In controlled experiments across five rotational speeds, the method achieved an accuracy of approximately 98.3% when trained on 75 operating cycles and evaluated over 15 cycles. The average precision and recall across all sensors exceeded 97% and 96%, respectively. The ability of the algorithm to treat microphone and accelerometer signals identically simplifies implementation, and the detection accuracy can be increased further by adding additional sensors.

## 1. Introduction

The development of systems enabling autonomous vessel navigation has recently attracted increasing interest [1,2]. One of the most important requirements in the development of autonomous vessels is the ability to monitor the proper operation of a propulsion engine. The significance of timely fault detection in engine operation primarily lies in increasing safety, preventing damage, and extending the engine lifespan, consequently reducing maintenance costs. Diesel engines are the most commonly used for propulsion systems and power supply in vessels [3]. Therefore, the ability to detect irregularities in the operation of diesel engines is of utmost importance to ensure the high reliability of the entire system.

Particularly interesting systems for detecting faults in engine operation are based on the analysis of vibration signals [4,5] or acoustic signals [6,7] produced by the engine. In previous research, various methods for detecting faults in diesel engine operation have been presented, which are based on the analysis of vibration or acoustic signals. The proposed methods include signal analysis in the frequency domain using fast Fourier transform (FFT) [8,9], signal analysis using wavelet transform (WT) [6,7,10,11,12], and signal decomposition methods such as empirical mode decomposition (EMD) [13,14], variational mode decomposition (VMD) [4,5], and principal component analysis (PCA) [6,15]. Recently, systems based on neural networks for detecting faults in ship systems have become increasingly interesting [5,16,17,18].

In [8], the authors investigated the possibility of monitoring the operation of internal combustion engines by analyzing the acoustic signal captured in the immediate vicinity of the engine. The possibility of detecting knocking, misfiring, and intake faults was considered. The proposed method, which is based on frequency analysis using FFT, enables the detection of faults in engine operation. The presented results do not include data on the reliability of the fault detection using the proposed method.

A time–frequency analysis of acoustic signals using wavelet packet transform (WPT) for feature extraction, based on which engine operation is classified, was presented in [6]. After feature extraction, three different approaches were compared: standard classification, Bayesian optimization, and a PCA method combined with Bayesian optimization. The proposed method enables reliable misfire detection; however, it requires a significant amount of time for training and testing.

In [19], a method for feature extraction based on the mel-frequency cepstrum (MFC) and VMD analysis of vibration signals was presented. The proposed method facilitates fault detection using the K-nearest neighbor classifier and was tested specifically on valve clearance faults. The proposed method is computationally demanding owing to feature sets with a large number of dimensions; thus, an improvement using vector quantization (VQ) was proposed to partially alleviate this issue.

Fault detection in engine operation using a combination of adaptive recursive variational mode decomposition (ARVMD) and component energy distribution spectrum (CEDS) on signals obtained from vibration measurements is proposed in [4]. ARVMD is used to extract intrinsic mode functions (IMFs), from which central frequencies and energies in unit frequency bands are obtained. Final classification is achieved by ranking the correlations of CEDS.

Detection of spectral anomalies using a variational autoencoder (VAE) is proposed in [17,20]. The proposed algorithm involves collecting data during both normal and faulty engine operations, feature extraction, and a training phase, in which a VAE is established and used for anomaly detection. The proposed algorithm enabled the detection of various faults with high reliability. Methods that use deep learning techniques require a sufficient amount of training data, which, in some situations, represents a significant drawback for the practical implementation of detection systems.

Beyond the aforementioned approaches, several advanced spectral techniques have been developed in other disciplines that may inspire improvements in engine fault detection. Least-squares wavelet analysis (LSWA) and its cross-wavelet extension compute time–frequency representations by directly fitting sinusoidal components to irregularly sampled data, providing instantaneous frequency estimates without pre-processing [21]. These methods have been applied to astronomical and interferometry time series and show strong performance in detecting anomalies and coupling between signals. Multichannel antileakage least-squares spectral analysis (MALLSSA) and the antileakage Fourier transform (ALFT) iteratively estimate and subtract dominant Fourier components from irregularly sampled hydrophone and seismic records to mitigate spectral leakage [22,23]. While offering improved spectral estimation over conventional FFTs, these methods are computationally intensive and require iterative optimization. Recent work on compressed sensing of vibration signals for rotating machinery faults constructed an order basis using randomly sampled rotational speed data; this sparse representation achieves up to twenty-fold compression and robust reconstruction under speed variations [24,25]. In [26], a novel wavelet-based spatiotemporal sparse quaternion dictionary learning (WSTS-QDL) method is proposed for the reconstruction of multi-channel vibration data. The approach exploits quaternion transforms for handling multi-dimensional channels, integrates wavelet decomposition for spatiotemporal feature extraction, and applies sparse dictionary learning to accurately reconstruct vibration signals. In [27], a deep learning-based sparsity-free compressive sensing method is developed for high-accuracy reconstruction of structural vibration responses. The proposed approach avoids the limitations of traditional sparsity assumptions by leveraging neural networks to directly learn the mapping between compressed measurements and full vibration signals, thereby improving reconstruction accuracy. In [28], a novel wind turbine fault diagnosis method is proposed that combines compressive sensing with a lightweight SqueezeNet model. Compressive sensing is employed to efficiently reduce the dimensionality of vibration data, while the SqueezeNet-based deep learning model enables accurate and computationally efficient fault classification.

To the best of our knowledge, such techniques have not yet been applied to marine diesel engine fault detection. Our work complements these developments by proposing a simpler FFT-based measure that can operate under varying engine speeds with minimal training data.

Although a substantial body of literature exists on diesel engine fault detection, several gaps remain. Many time–frequency and machine learning approaches rely on extensive labeled data and are sensitive to engine speed, while recent spectral methods such as LSWA, MALLSSA, ALFT, and compressed sensing provide improved frequency estimation or sparse representations, yet have been applied primarily in astronomy, geophysics, and bearings diagnostics. None of these techniques have been explored for marine diesel engines, and there is a lack of simple methods that can simultaneously handle acoustic and vibration measurements across variable speeds.

The main contributions of this paper are summarized as follows:Novel frequency-domain fault measure: We introduce a simple metric based on the ratio of DFT magnitude spectra obtained from monitoring and training data and use the two largest spectral peaks to compute a distance that distinguishes normal from faulty operation.Unified processing of acoustic and vibration signals: The proposed algorithm applies identically to microphone and accelerometer signals, demonstrating comparable detection performance for both modalities and highlighting the universality of the approach.Comprehensive experimental evaluation: We validate the method on a six-cylinder marine diesel engine at five rotational speeds, emulate faults by disabling individual cylinders, and report detailed performance metrics including accuracy, precision, recall, and F1-score.Parameter analysis and practical guidelines: The influence of window functions, FFT frame length, the number of cycles used for training and detection, and the threshold scaling parameter is systematically analyzed, providing guidance for practitioners.

The work presented in this paper is based on our previous work already published in [29]. Our previous paper described the effect observed during the research, specifically, that a vector could be defined, from which we analyzed the distribution of the elements to classify engine operation. In addition, the basic principle of the algorithm operation was presented. In this work, the measure and threshold for classification are defined and elaborated upon in detail, along with the training process, and the performance of the proposed algorithm is thoroughly presented.

This paper presents an algorithm that enables the classification of motor operations as either correct or faulty by analyzing the signals obtained from a microphone or accelerometer. The algorithm is based on signal analysis in the frequency domain. The recognition process includes a training phase followed by detection. Training was performed by analyzing the signals over a certain number of full diesel engine operating cycles. Training and detection could be conducted at any engine speed. The classification was performed based on a threshold, the value of which was determined during the training phase. A key advantage of the proposed algorithm is its simplicity and ability to detect operating faults by applying the same algorithm to signals obtained from either a microphone or an accelerometer. In addition, the reliability of detection can be increased by adding more microphones or accelerometers. Particular emphasis can be placed on the simplicity of practical implementation. This method was tested using a marine diesel engine using acoustic and vibration signals. The proposed algorithm successfully classifies engine operation as either normal or faulty based on acoustic and vibration signals at different engine speeds. A motor fault was emulated by deactivating one cylinder.

The remainder of this paper is organized as follows. Section 2 describes the materials and methods. The results are presented in Section 3, followed by a detailed discussion in Section 4. Finally, Section 5 provides the conclusions of this study.

## 2. Materials and Method

### 2.1. Datasets

Measurements were conducted on a six-cylinder, four-stroke marine diesel engine installed on a working vessel. Two microphones were placed near the engine and four accelerometers were mounted on the engine housing, while an optical tachometer sensor was attached to the flywheel to register crankshaft revolutions. The signals from these sensors were synchronously digitized using a 24-bit A/D converter at a sampling rate of 51.2 kHz. Data were collected at five rotational speeds (600, 900, 1200, 1500, and 1800 RPM) for at least 30 s per speed under normal operation. Faulty operation was emulated by deactivating either the first or the fifth cylinder, producing two independent datasets. Each dataset consisted of seven synchronized channels (two microphones, four accelerometers, and one tachometer) used for training and testing, as described below.

### 2.2. Method

The method proposed in this paper enables fault detection in the operation of a diesel engine by processing signals obtained from a microphone and accelerometer. The system involves placing a microphone in close proximity to the engine and accelerometers on the engine housing. The signal from the microphone and accelerometer outputs were converted to digital form using an analog-to-digital (A/D) converter and processed on a computer. Considering that diesel engine operation is characterized by regular cycles that are related to the rotation of the engine itself, it was necessary to synchronize signal processing with the engine’s operating cycles. For this purpose, an optical tacho sensor was used to register one full revolution of the engine crankshaft. The optical tacho sensor was placed in the immediate vicinity of the flywheel and registered one full turn, providing a rectangular pulse at the output. Figure 1 shows the signal acquired at the output of the optical tacho sensor during the engine operation. Given that the measurements were performed on a four-stroke diesel engine, two revolutions corresponded to one full engine operation cycle. A voltage level of 3 V was taken as the threshold, indicating the beginning of a new full cycle on every second positive pulse edge.

Let $x_{s}\left[n\right]$ denote the discrete time-domain signal sampled from the output of the microphone or accelerometer at discrete times $t=nT_{s},n=0,1,\ldots$, where $T_{s}$ is the reciprocal value of the sampling frequency $f_{s}$, that is, $T_{s}=1/f_{s}$. Using the signal from the optical tachometer sensor output $x_{T}\left[n\right]$, a vector $x_{s}\left[m\right]$ is defined, with elements that are samples of the signal $x_{s}\left[n\right]$ corresponding to the *m*th full cycle of engine operation:(1)$$x_{s}\left[m\right]={\left[x_{s}\left[g_{X_{T}}\left(m\right)\right],\ldots ,x_{s}\left[g_{X_{T}}\left(m\right)+N_{m}^{\prime }-1\right]\right]}^{⊤},$$
where $g_{X_{T}}:m\to n$ denotes the function that maps the full cycle index *m* to index *n*, with which the *m*th full cycle begins, determined by every second positive edge of the signal $x_{T}\left[n\right]$, $N_{m}^{\prime }$, which denotes the number of samples collected during the *m*th full cycle of engine operation. ${\left[\cdot \right]}^{⊤}$ stands for the transposition operator. Given that the sampling frequency is constant, the number of samples collected during one full cycle $N_{m}^{\prime }$ depends on the rotational speed of the engine. Considering that the engine classification algorithm should operate under varying rotational speeds, it is necessary to resample the signal $x_{s}\left[m\right]$ to obtain a signal that has the same number of samples per one full cycle for any rotational speed of the engine. By resampling the signal $x_{s}\left[m\right]$, a vector $x_{r}\left[m\right]$ is obtained as(2)$$x_{r}\left[m\right]={\left[x_{r}\left[0,m\right],x_{r}\left[1,m\right],\ldots ,x_{r}\left[N-1,m\right]\right]}^{⊤},$$
where *N* denotes the number of samples. We resampled the signal using the resample() function in MATLAB version 9.12 (R2022a), which performs polyphase anti-alias filtering followed by upsampling and downsampling to yield a band-limited interpolation. This step ensures that each cycle is represented by the same number of samples across different rotational speeds while minimizing spectral distortion. We also experimented with linear and spline interpolation; however, the polyphase approach offered the best compromise between computational cost and frequency-domain fidelity.

The proposed method includes the estimation of the frequency content of the signal being analyzed; therefore, the window function in the time domain [30] is used to improve the frequency resolution and, consequently, the accuracy of the fault detection. Experimenting with the measured data, it was observed that the reliability of the estimation of the presence of an error in the operation of the engine depended on the applied window function. By multiplying the samples of signal $x_{r}\left[\cdot \right]$ with the corresponding coefficients of the window function $w={\left[w\left[0\right],\ldots ,w\left[N-1\right]\right]}^{⊤}$, the signal $x_{w}\left[m\right]={\left[x_{w}\left[0,m\right],x_{w}\left[1,m\right],\ldots ,x_{w}\left[N-1,m\right]\right]}^{⊤}$ is obtained:(3)$$x_{w}\left[m\right]=x_{r}\left[m\right]⊙w,$$
where ⊙ denotes the Hadamard matrix product. After windowing, the DFT of signal $x_{w}\left[m\right]$ is calculated, and the DFT coefficients $X_{w}\left[k,m\right]$ are obtained. After applying the window function, the DFT of the signal $x_{w}\left[m\right]$ is computed, resulting in the DFT coefficients $X_{w}\left[k,m\right]$:(4)$$X_{w}\left[k,m\right]=\sum_{n=0}^{N-1}x_{w}\left[n,m\right]e^{-j\frac{2\pi nk}{N}},\;\forall \;k:0\leq k\leq N-1.$$

In the analysis of the signal spectrum, only the magnitudes of the DFT coefficient matter—phase information is not used. Therefore, we consider the absolute values $|X_{w}\left[k,m\right]|$ of the DFT rather than the complex coefficients themselves. Because the signal samples $x_{w}\left[n,m\right]$ are real-valued, it is sufficient to retain only the first N/2 magnitudes owing to the symmetry of the spectrum. A vector $X_{\left|w\right|}\left[m\right]$ is defined, whose elements are these magnitudes:(5)$$X_{\left|w\right|}\left[m\right]=\left[\right|X_{w}\left[0,m\right]|,|X_{w}\left[1,m\right]|,\ldots ,|X_{w}\left[N/2-1,m\right]{\left|\right]}^{⊤}.$$

The absolute values of the DFT coefficients $|X_{w}\left[k,m\right]|$ were calculated from the samples of the signal $x_{w}\left[m\right]$ acquired during the *m*th full cycle of engine operation. A more reliable estimate of these values can be obtained by computing the coefficients from multiple consecutive full cycles of engine operation:(6)$${\bar{X}}_{\left|w\right|}=\sum_{m=0}^{N_{C}-1}X_{\left|w\right|}\left[m\right],$$
where $N_{C}$ denotes the number of full cycles of engine operation. From the values of vector ${\bar{X}}_{\left|w\right|}$, a metric can be established to classify engine operation as either correct or faulty. Since the values in the vector ${\bar{X}}_{\left|w\right|}$ scale with the number of averages $N_{C}$, normalization is required to ensure consistency across different averaging settings. Therefore, vector ${\bar{X}}_{\left|w\right|}$ is normalized in such a way that its norm is equal to 1:(7)$${\bar{X}}_{n}=\frac{{\bar{X}}_{\left|w\right|}}{\left|\right|{\bar{X}}_{\left|w\right|}{\left|\right|}_{2}},$$
where $\left|\right|\cdot {\left|\right|}_{2}$ denotes the $L^{2}$ norm.

In the following text, the notation for the vectors ${\bar{X}}_{n}^{\left(M\right)}$ and ${\bar{X}}_{n}^{\left(T\right)}$ is introduced, which are calculated according to expression (Equation 7), where ${\bar{X}}_{n}^{\left(M\right)}$ refers to the vector computed from signal samples collected during normal operation (monitoring phase), while ${\bar{X}}_{n}^{\left(T\right)}$ is computed from samples collected during the training phase. Furthermore, the notations $N_{M}$ and $N_{T}$ are introduced to distinguish the number of complete engine operation cycles $N_{C}$ during normal engine operation and the training phase, respectively, which are used for the calculation in expression (6) in the corresponding case.

By analyzing the data obtained through measurements, we found that faults in engine operation can be detected from the ratio of the values of elements in vectors ${\bar{X}}_{n}^{\left(M\right)}=\left[{\bar{X}}_{n}^{\left(M\right)}\left[0\right],\ldots ,{\bar{X}}_{n}^{\left(M\right)}\left[N/2-1\right]\right]$ and ${\bar{X}}_{n}^{\left(T\right)}=\left[{\bar{X}}_{n}^{\left(T\right)}\left[0\right],\ldots ,{\bar{X}}_{n}^{\left(T\right)}\left[N/2-1\right]\right]$:(8)$$X_{F}={\bar{X}}_{n}^{\left(M\right)}⊘{\bar{X}}_{n}^{\left(T\right)},$$
where ${\bar{X}}_{n}^{\left(M\right)}$ denotes the vector calculated using Equation (Equation 7) during normal operation, which can be with or without failure; ${\bar{X}}_{n}^{\left(T\right)}$ denotes the vector calculated using Equation (Equation 7) during operation without failure, i.e., the training phase; and ⊘ denotes the Hadamard division. Furthermore, let $X_{F_{1}}$ and $X_{F_{2}}$ denote the largest and next-largest values of the elements in the vector $X_{F}$, respectively. Through our research, we concluded that the values of $X_{F_{1}}$ and $X_{F_{2}}$ change significantly if there is a change in the operation of the motor; in other words, a change in the engine operation can be identified by calculating the $L^{2}$ norm of these values:(9)$$d=\sqrt{X_{F_{1}}^{2}+X_{F_{2}}^{2}}.$$

The fault detection process includes a training phase, during which proper operation of the engine must be ensured. At a specific engine rotational speed, the vector of reference values ${\bar{X}}_{n}^{\left(T\right)}={\bar{X}}_{n}$ is calculated for each of the signals obtained from the microphone and accelerometer during the training phase over $N_{C}=N_{T}$ full cycles of engine operation using Equation (Equation 7). The training phase must be carried out at all different engine rotation speeds at which faulty operation detection will be performed. Upon completion of the training phase, the monitoring phase commences, during which the vector ${\bar{X}}_{n}^{\left(M\right)}={\bar{X}}_{n}$—also computed using Equation (Equation 7)—is calculated over $N_{M}$ full cycles of engine operation. Subsequently, the vector $X_{F}$ is calculated using Equation (Equation 8), where two peak values are identified, from which *d* is then calculated using Equation (Equation 9). The procedure is repeated for each of the following $N_{C}=N_{M}$ full cycles during the operation of the engine. Applying the proposed algorithm to signals obtained from the microphones and accelerometers, it is demonstrated that the value of variable *d* depends on whether there has been a change in the engine’s operating mode compared to the mode during the training phase. The classification of engine operation as either correct or faulty can be determined by comparing the value of *d* obtained during the monitoring phase of operation with the threshold value $d_{T}$ determined during the training phase.

## 3. Results

Measurements were conducted in a ship’s engine room on a four-stroke diesel engine with six cylinders. The engine displacement per cylinder was 4.88 L, the power output was 2525 kW, and the maximum rotational speed was 1900 revolutions per minute (RPM). Two microphones were placed in close proximity to the engine and four accelerometers were mounted on the engine housing. In addition, an optical tachometer sensor was positioned to register the complete rotation of the flywheel. The engine room with the measuring equipment, microphone, and accelerometer is shown in Figure 2. The signals obtained from the outputs of the microphones, accelerometers, and optical tachometer sensor, totaling seven channels, were digitized using an analog-to-digital (A/D) converter with a sampling frequency of $f_{s}$ = 51,200 Hz. The resolution of the A/D converter was 24 bits per sample. The sampling in all the channels was synchronized. The measurements of the sound signal and vibrations were conducted on a properly functioning engine at 600, 900, 1200, 1500, and 1800 RPM. Faulty engine operation was simulated by disabling one cylinder. Two independent measurements were conducted with the first or fifth cylinder deactivated.

The performance of the proposed algorithm is shown in Figure 3 and Figure 4, respectively. Figure 3 displays the results for engine operation at 600, 900, and 1200 RPM (from left to right), while Figure 4 shows the results for operation at 1500 and 1800 RPM, also from left to right. To calculate expression (Equation 3), in the time domain, a Hamming window function was used with a sample size of N=4096. The presented results pertain to a measurement scenario in which the outcomes of proper engine operation are compared with the results obtained when the first cylinder was deactivated. Hereafter, the term ’block’ will be used to denote multiple consecutive full cycles of engine operation. For each block of $N_{C}=N_{M}=20$ full cycles, the value *d* was calculated according to Equation (Equation 9), after which the average value ${\bar{d}}_{M}$ was computed. Let $d_{i}$ denote the value obtained for the *i*-th block; it follows that ${\bar{d}}_{M}=1/N_{E}\sum_{i=1}^{N_{E}}d_{i}$, where $N_{E}$ denotes the number of blocks of the collected data for each microphone or accelerometer at different revolution speeds during normal and faulty operations. Owing to limited measurement capabilities, the duration of the collected audio recordings was at least 30 s for each engine revolution count for which the measurements were conducted. A specific challenge was encountered during data collection when one cylinder was deactivated, given the potential damage that could occur if the engine operated for an extended period with a deactivated cylinder. The number of complete cycles in the specified time interval being measured, $N_{E}$, varied from around 15 to 80. The blue bars in Figure 3 and Figure 4 represent the values of ${\bar{d}}_{M}$ obtained from data collected during normal operation, while the orange bars correspond to the data collected during faulty operation. Faulty operation refers to a situation in which the first cylinder was disabled. Also, on the graphs for each individual measurement, the double deviation for each average value of ${\bar{d}}_{M}$ is depicted for normal as well as faulty engine operation. Specifically, the mean value ${\bar{d}}_{M}$ is shown along with the double standard deviation ${\bar{d}}_{M}\pm 2\sigma_{d}$. In all cases, the vector ${\bar{X}}_{n}^{\left(T\right)}={\bar{X}}_{n}$ was calculated from $N_{C}=N_{T}=100$ full cycles of engine operation, using data collected during the training phase. From the obtained results depicted in Figure 3 and Figure 4, a clear difference in the mean value of the measure ${\bar{d}}_{M}$ was evident between proper engine operation and faulty engine operation, considering various sensors at different engine speeds. A weaker detection capability is observed in Figure 3 at 900 RPM, where the classification threshold was situated within the deviation range of $\pm 2\sigma_{d}$ from the mean value of ${\bar{d}}_{M}$. Independent data were used for training for all tested rotational speeds and for all sensors. The training data were not used to test the method. Figure 3 and Figure 4 demonstrate that the value of ${\bar{d}}_{M}$ was significantly influenced by whether the engine operated with all cylinders active or with the first cylinder deactivated. This pattern was consistent across all tested engine rotational speeds, both for microphones and accelerometers.

To classify the mode of operation as either correct or faulty, it was necessary to determine a threshold. By analyzing the collected data, we concluded that the threshold could be determined from the training data, from which vector ${\bar{X}}_{n}^{\left(T\right)}$ was also calculated. The procedure for threshold determination was as follows: data collected during all full cycles used for training, 100 in the case of the results shown in Figure 3 and Figure 4, were divided into 20 blocks, each consisting of five full cycles of engine operation. For each block of $N_{C}=N_{M}=20$ full cycles, the value *d* was calculated according to Equation (Equation 9), after which the average value $d_{T}$ was computed. Let $d_{i}$ denote the value obtained for the *k*-th block; it follows that $d_{T}=1/N_{M}\sum_{k=1}^{N_{M}}d_{k}$. The parameter $d_{T}$ was crucial for determining the threshold to be used to classify engine operation as either correct or faulty.

Through experimental analysis, we determined that the ‘optimal threshold’ for classifying engine operation could be defined using the parameter $d_{T}$, according to the criterion of maximizing classification accuracy (Acc):(10)$$‘\mathrm{optimal}\;\mathrm{threshold}’=d_{T}\cdot \arg\max_{C}\mathrm{Acc},$$
where *C* is a parameter for optimization and Acc is defined as(11)$$\mathrm{Acc}=\frac{\mathrm{TP}+\mathrm{TN}}{\mathrm{TP}+\mathrm{TN}+\mathrm{FP}+\mathrm{FN}}\cdot 100\;\left[\%\right],$$
where TP, TN, FP, and FN denote the true positive, true negative, false positive, and false negative test results, respectively. TP, TN, FP, and FN refer to the total number of events determined from the collected data from each microphone and accelerometer, respectively. In Figure 3 and Figure 4, the horizontal line on the bar graphs represents a threshold of $1.2\cdot d_{T}$, i.e., C=1.2. Further details on how the parameter C=1.2 was determined are provided in the explanation of the results shown in Figure 5. A more reliable estimate of the threshold can be obtained by assessing it from a training dataset that is not used for calculating the vector ${\bar{X}}_{n}^{\left(T\right)}$. In such cases, a longer training sequence is required. However, the obtained results justify the proposed approach. Algorithm 1 presents the proposed method.

Figure 5 shows the accuracy achieved by the proposed algorithm as a function of the parameter *C*, which, when multiplied by $d_{T}$, defines the threshold value used for classifying engine operation as either correct or faulty.  
**Algorithm 1** Algorithm for training and detecting engine operating faults.$N\leftarrow 4096,N_{T}\leftarrow 75,N_{M}\leftarrow 15$w←window(‘Hamming’,N)$N_{T}^{\prime }\leftarrow \left\lfloor N_{T}/5\right\rfloor$counter←0,m←0$N_{C}\leftarrow N_{T}$${\bar{X}}_{\left|w\right|}^{\prime }\left[:,:\right]\leftarrow 0_{N/2\times N_{T}^{\prime }}$Training←TRUE**while** monitoring continues **do**     m←m+1     $N_{m}^{\prime }\leftarrow$ number of samples within *m*th full cycle     ${\bar{X}}_{\left|w\right|}\leftarrow 0_{N/2\times 1}$     **for** $m^{\prime }\leftarrow 0,N_{C}-1$ **do**           $x_{s}\leftarrow {\left[x_{s}\left[g_{X_{T}}\left(m\right)\right],\ldots ,x_{s}\left[g_{X_{T}}\left(m\right)+N_{m}^{\prime }-1\right]\right]}^{⊤}$           $x_{r}\leftarrow \mathrm{interp}\left(x_{s},N\right)$           $x_{w}=x_{r}⊙w$           $X_{w}=\mathrm{FFT}\left(x_{w}\right)$           $X_{\left|w\right|}\leftarrow \left|X_{w}\left[k\right]\right|,\;k=0,\ldots ,N/2-1$           ${\bar{X}}_{\left|w\right|}\leftarrow {\bar{X}}_{\left|w\right|}+X_{\left|w\right|}$           **if** Training==TRUE **then**                ${\bar{X}}_{\left|w\right|}^{\prime }\left[:,counter\right]\leftarrow {\bar{X}}_{\left|w\right|}^{\prime }\left[:,counter\right]+X_{\left|w\right|}$                $counter\leftarrow counter+\lfloor m^{\prime }/5\rfloor$          **end if**    **end for**    ${\bar{X}}_{n}\leftarrow {\bar{X}}_{\left|w\right|}/\left|\right|{\bar{X}}_{\left|w\right|}{\left|\right|}_{2}$    **if** Training==TRUE **then**          ${\bar{X}}_{n}^{\left(T\right)}\leftarrow {\bar{X}}_{n}$          $d_{T}\leftarrow 0$          **for** $k\leftarrow 0,N_{T}^{\prime }-1$ **do**                ${\bar{X}}_{n}\leftarrow {\bar{X}}_{\left|w\right|}^{\prime }\left[:,k\right]/\left|\right|{\bar{X}}_{\left|w\right|}^{\prime }\left[:,k\right]{\left|\right|}_{2}$                $d_{T}\leftarrow d_{T}+$
calculate-d
$\left({\bar{X}}_{n},{\bar{X}}_{n}^{\left(T\right)}\right)$          **end for**          $d_{T}\leftarrow 1.2\cdot d_{T}/N_{T}^{\prime }$          Training←FALSE          $N_{C}=N_{M}$    **else**          ${\bar{X}}_{n}^{\left(M\right)}\leftarrow {\bar{X}}_{n}$          d=
 calculate-d
$\left({\bar{X}}_{n}^{\left(M\right)},{\bar{X}}_{n}^{\left(T\right)}\right)$          **if** $d\leq d_{T}$ **then**               display(‘Normaloperation’)          **else**               display(‘Failureinoperation’)          **end if**      **end if****end while****function** calculate-d(${\bar{X}}_{n},{\bar{X}}_{n}^{\left(T\right)}$)      $X_{F}={\bar{X}}_{n}⊘{\bar{X}}_{n}^{\left(T\right)}$      $X_{F1},X_{F2}\leftarrow$ findPeaks($X_{F}$)      $d=\sqrt{\left(X_{F1}^{2}+X_{F2}^{2}\right)}$     **return** *d***end function****function** findPeaks($X_{F}$)      $\mathrm{Find}\;\mathrm{two}\;\mathrm{peaks}\;\mathrm{from}\;X_{F},\mathrm{not}\;\mathrm{closer}\;\mathrm{than}\;10\;\mathrm{samples}$      **return** $X_{F1},X_{F2}$**end function**

The images, from left to right, correspond to $N_{T}$ = 50, 75, and 100 full cycles used for training. From these, 10, 15, and 20 full cycles, respectively, were used to form a block, from which the $d_{k}$ values were calculated. When averaged, these yielded the $d_{T}$ value. Each image shows graphs depicting the dependence of classification accuracy on the parameter *C*. Individual graphs correspond to the performances achieved by analyzing data for method validation, using $N_{M}$ = 5, 10, 15, and 20 full cycles to calculate the *d* value using Equation (Equation 9). Based on this value, the operation was classified as correct or faulty, depending on whether it was below or above the $d_{T}$. A Hamming window function was used with a sample size of N=4096, and the obtained results correspond to the scenario in which faulty engine operation was simulated by deactivating the fifth cylinder. The choice of the parameter *C* was highly important as the accuracy of the classification depended on it. Analyzing the obtained results, it can be observed that for $N_{T}=50$ (left image), the maximum accuracy was achieved when *C* was chosen in the range of 1.35 to 1.45, depending on the value of $N_{M}$. However, since the maximum achievable accuracy for $N_{T}=50$ was approximately 0.92, this case is not particularly interesting given that significantly better accuracy values were obtained by increasing the number of full cycles required for training, as can be seen in the middle and right images. In the case when $N_{T}=75$ and 100 (middle and right images), the highest accuracy was achieved when *C* was in the range of 1.2 to 1.25 and when $N_{M}=15$ or 20. From the obtained results, it can be concluded that for the values of variables $N_{T}$ and $N_{M}$ relevant for practical use, namely, 75 and 100 for $N_{T}$ and 15 and 20 for $N_{M}$, maximum accuracy was achieved when *C* was in the range of 1.2 to 1.25.

Figure 6 and Figure 7 depict the dependence of accuracy on the number of full cycles, $N_{M}$, used to calculate the value of *d* for different values of full cycles used for training, $N_{T}$, in the case of simulating errors in operation by deactivating the first and fifth cylinder, respectively. To calculate the threshold, the optimal parameter C=1.2 was used, which maximized accuracy. According to Figure 5, for practical values ($N_{T}=75$ or 100, $N_{M}=15$ or 20), the maximum accuracy was achieved when *C* lay between 1.2 and 1.25. Therefore, C=1.2 was selected as a representative near-optimal value. A Hamming window was applied, and N=4096.

From the results, it is evident that the number of full cycles required to calculate the value of *d*, which determined the threshold for classification, affected the detection accuracy. With $N_{M}=15$ full cycles, an accuracy of approximately 98% was achieved, while increasing $N_{M}$ to 20 yielded no significant improvement in accuracy when $N_{T}=75$. It was even slightly reduced when the error was simulated by deactivating the first cylinder. A smaller number of full cycles of engine operation required for classification, $N_{M}$, is desirable because it requires less time to determine whether an error has occurred. This is particularly important at lower engine speeds because there are fewer full cycles per unit time, which prolongs the time needed to collect data for assessment. For example, at 600 RPM, 15 full cycles take 3 s, while, at 1800 RPM, 15 full cycles take 1 s.

From the displayed results, it is particularly interesting that higher accuracy was achieved when training was conducted using $N_{T}=75$ full cycles of engine operation compared to when $N_{T}=100$. As with $N_{M}$, it is preferable for training to require a smaller number of full cycles of engine operation, which means shorter training times. For example, at 600 RPM, 75 full cycles take 15 s, while, at 1800 RPM, 75 full cycles take 5 s.

The displayed results show that the achieved accuracy was similar when faulty operation was emulated by deactivating the first cylinder compared to deactivating the fifth cylinder. Considering the obtained accuracy, it can be concluded that the optimal choice was $N_{T}=75$ and $N_{M}=15$. Unfortunately, owing to objective circumstances, we were not able to explore the possibility of detecting other types of potential faults in engine operation as the measurements were conducted on an engine in commercial operation, and the risk of any damage was not acceptable.

Table 1 provides data on the total number of events (TP, FP, FN, and TN) collected from all sensors, along with the corresponding accuracies observed for all measurements, conducted in the case of deactivation of the first or fifth cylinders.

In Figure 8, the classification success based on the data collected from all microphones and accelerometers is graphically illustrated for all engine speeds tested during normal engine operation and operation when the first or fifth cylinder was deactivated. The results presented correspond to the scenario where $N_{T}$ = 50 and $N_{M}$ = 5, using the Hamming window function and *N* = 4096. The incorrect classification is marked with a red dot, whereas the correct one is marked with a green dot. The presented results do not correspond to the scenario in which $N_{M}$ was chosen to achieve the maximum accuracy. The purpose of the illustration is to provide a good insight into the classification potential at different engine speeds for all microphones and accelerometers. From the obtained results, it can be seen that at certain engine speeds, in this case, 900 RPM, the classification was not satisfactory during normal engine operation because it was incorrect in a large number of cases. It can also be observed that when the fifth cylinder was deactivated, the classification was incorrect in the majority of cases for the data collected from accelerometer Acc 2 at 1200 RPM. Similar observations can be made for Mic 1 at 900 and 1500 RPM and Mic 2 at 1500 and 1800 RPM. It should be noted that the classification was often unsuccessful for data collected from a particular microphone or accelerometer, but, under the same conditions, it was successful for data collected from other microphones or accelerometers. Hence, the principle of classification based on the majority could be applied.

In the case of the first cylinder being turned off in only two cases, indicated by a red vertical line at 600 RPM, from the data collected from two microphones and one accelerometer (Mic 1, Mic 2, and Accl 1), the faulty operation was classified as correct, while, using the data collected from the remaining three accelerometers (Accl 2, Accl 3, and Accl 4), the classification was correct. In the case of the fifth cylinder being turned off, it can be observed that the classification would be correct in every instance as, at most, two out of six microphones or accelerometers had incorrect classifications simultaneously.

By applying the same classification principle with $N_{T}=75$ and $N_{M}=15$, for which the maximum precision was achieved, the classification was incorrect in only one case in which correct engine operation using data collected from both microphones (Mic1 and Mic2) and one accelerometer (Acc3) was classified as faulty, as shown in Figure 9 by the red vertical line, whereas the remaining three accelerometers were classified as correct. In all other cases, for both correct and faulty operations, regardless of whether the first or fifth cylinder was disabled, the classification was correct.

### Additional Performance Metrics

While accuracy provides an overall measure of correct classifications, other metrics such as precision, recall, and F1-score are informative when dealing with imbalanced data [31]. Precision reflected the proportion of detections that were actually faults, recall indicated the proportion of actual faults that were correctly detected, and the F1-score was their harmonic mean. Table 2 summarizes these metrics for the representative case where $N_{T}=75$ and $N_{M}=15$ for both the first and fifth cylinder faults. The values were computed from the confusion matrix entries in Table 1.

As described in Section 2.1, the proposed method for classifying engine operation is based on calculating the value of *d* according to Equation (Equation 9), for which it is necessary to determine the two largest values in the vector $X_{F}$ defined in Equation (Equation 8). In Figure 10, examples of the highest values of elements in vector $X_{F}$ are shown for correct engine operation in the left column images and for faulty operation with the first cylinder turned off in the right column images at different engine speeds. The displayed images correspond to the case when $N_{T}=75$, $N_{M}=15$, and N=4096 and the Hamming window function was applied. The red dots indicate the highest values of the samples selected according to the proposed algorithm. In some cases, the largest values were clustered around a certain position *n*; in such cases, the next largest value from that group was not considered. This example is illustrated in the image depicting the correct engine operation at 1500 RPM and the faulty engine operation at 900 RPM. To avoid selecting the neighboring maximum values, the condition for choosing the second maximum value was that it needed to be at least ten positions away from the position of the maximum value. In the displayed images, it can be observed that the maximum values of the signal samples were lower in the case of normal motor operation compared to the values for faulty motor operation, allowing for classification.

By analyzing the obtained results, we determined that the accuracy of the proposed method also depends on the applied window function. Table 3 presents the achieved accuracy using different window functions. The results pertain to the case where there was no fault in operation and when the first cylinder was excluded, with $N_{T}=75$, $N_{M}\;=\;15$, and N=4096. The obtained results show that the use of a window function is justified. The poorest result was obtained when no windowing function was applied, i.e., when it was rectangular, while the best result was achieved when using the Hamming window functions.

The achieved accuracy also depended on the number of samples used to calculate FFT. Table 4 provides an overview of the achieved detection accuracy for disabling the first or fifth cylinder for different numbers of sample frames in the FFT. The results indicate that the efficiency decreased as expected, but not significantly, with a reduction in the number of samples. This suggests that the upper frequency limit covered in the signal analysis is not critical for the success of error detection in engine operation. The most critical situation occurred when the motor spun at its slowest speed, which, in this case, was 600 RPM. Given that the sampled signal $x_{s}\left[n\right]$ is resampled at *N* samples within two full cycles of motor operation, sampling frequency after resampling can be expressed as(12)$$f_{s^{\prime }}=\frac{f_{\mathrm{eng}}N}{2},$$
where $f_{\mathrm{eng}}$ represents the number of revolutions of the engine per unit of time. The division by a factor of 2 is because a full cycle of engine operation involves two revolutions. In the case where the rotation speed was 600 RPM and for N=128 signal samples, according to Equation (Equation 12), the sampling frequency after resampling is $f_{s^{\prime }}=640$ Hz. It follows that the maximum frequency covered by the analysis was $f_{s^{\prime }}/2=320$ Hz. Regardless of the fact that the frequency content of the signal was analyzed up to a maximum of 320 Hz, the accuracy of fault detection was relatively high, approximately 95%. This was because the proposed method for detecting engine faults is based on finding the two maximum values of the samples in vector $X_{F}$ and calculating the value of *d* according to Equation (Equation 9). By analyzing the distribution of the signal samples $X_{F}$, it was observed that the distribution differed between normal motor operation and faulty motor operation. In Figure 11, histograms of the signal samples $X_{F}$ are depicted at 600 and 1800 RPM for N=128 and N=4096 in both normal and faulty motor operations. From the displayed histograms, it can be observed that the maximum values of the signal samples $X_{F}$ were higher in the case of faulty motor operation compared to those for normal operation. This held true for different rotational speeds and values of *N*. The results are presented for 600 and 1800 RPM and N=128 and N=4096 for simplicity; however, from other conducted experiments and displayed results, it can be concluded that this holds true in general. It can be concluded that for the success of detecting faulty motor operation using the proposed method, the distribution of signal $X_{F}$, i.e., the presence of higher maximum values of signal samples in the case of faulty motor operation compared to normal operation, is crucial.

Figure 12 shows an example of the distribution of variable *d* during correct and incorrect engine operations at 1800 RPM. The figure illustrates the difference in the distribution of variable *d* between correct and incorrect engine operations. Specifically, the values of variable *d* are higher during incorrect engine operation than during correct engine operation, which enables classification.

## 4. Discussion

### 4.1. Comparison with Alternative Spectral Methods

Least-squares wavelet analysis [21] and its cross-wavelet extension provide high-resolution time–frequency and phase estimates on irregular grids, while MALLSSA and ALFT iteratively remove dominant Fourier components to mitigate spectral leakage [22,23]. Compressed sensing with order bases yields sparse representations for rotating machinery under speed variations. These methods are more computationally demanding and have been applied primarily in astronomy, seismology, and bearing fault diagnosis. In contrast, our FFT-based distance measure is simple, operates on regularly sampled data, and can be implemented on embedded hardware. Table 5 shows accuracies for various methods dealing with the problem of engine fault detection and classification. The proposed method achieves accuracy comparable with competing approaches, but it relies on FFT, making it computationally less demanding in the training and testing phases. It should be noted that completely objective comparison with the proposed method is not feasible because the competing approaches deal with different types of engine failures (engine misfires, insufficient oil for fuel supply, etc.), while our algorithm focuses on the detection of situations where the first or fifth cylinder is disabled. Also, test setups vary significantly depending on the engine type and environment.

### 4.2. Uncertainties and Limitations

The experiments were performed on a single vessel and only one type of fault (cylinder deactivation) was emulated. Ambient noise, mechanical variability and operational disturbances may affect spectral features. Although window functions reduce spectral leakage, uncertainties arise from the choice of window and the number of cycles used for averaging. Future studies should investigate robustness under artificially added noise and other fault types (valve clearance faults, fuel injection anomalies, bearing wear) and assess the impact of different window functions and interpolation methods.

### 4.3. Necessity of Combining Acoustic and Vibration Measurements

Our results show that individual sensors sometimes misclassify at certain speeds. Microphones capture radiated sound whereas accelerometers measure structural vibrations; their combination provides complementary information. A majority vote strategy across multiple microphones and accelerometers mitigates misclassifications from individual sensors and improves robustness.

### 4.4. Advantages and Future Work

The proposed method requires only a few seconds of data for training and detection and can operate in real time with modest computational resources. It is transparent and does not rely on black-box models, making it suitable for safety-critical marine applications. Future work will evaluate performance in noisy and transient conditions; compare the method with LSWA, MALLSSA, ALFT, and compressed sensing on the same dataset; investigate adaptive thresholds and multi-sensor fusion strategies; and extend the approach to detect a broader range of faults.

## 5. Conclusions

This paper presents a simple method for detecting faults in the operation of a marine diesel engine by analyzing acoustic or vibration signals. The method is based on a frequency analysis of the signals using FFT and defines a distance measure that classifies engine operation as either correct or faulty. We validated the method by analyzing acoustic and vibration signals obtained from a marine diesel engine at different rotation speeds while emulating faults by disabling the first or fifth cylinder. The achieved accuracy in the representative case ($N_{T}=75$, $N_{M}=15$) was approximately 98.3%, with precision and recall above 97% and 96%, respectively. The accuracy of the proposed method is comparable with the state-of-the-art methods for engine fault diagnosis. A key advantage of the method is its ease of training, which does not require a large number of cycles of correct engine operation, allowing a reference database to be built quickly for different operating conditions. The proposed method allows for the detection of faults from only 15 full cycles of engine operation, and good accuracy can be achieved even with a small number of FFT frame samples, enabling implementation on simple computational resources. Although similar detection performance was obtained for both microphones and accelerometers, the universality of the method needs to be confirmed through further tests with other fault types and different sensor configurations. An additional practical advantage is that implementation requires no prior preparation of the engine: it is sufficient to place microphones in close proximity to the engine and accelerometers on the engine housing.

## Figures and Tables

**Figure 1 sensors-25-05669-f001:**
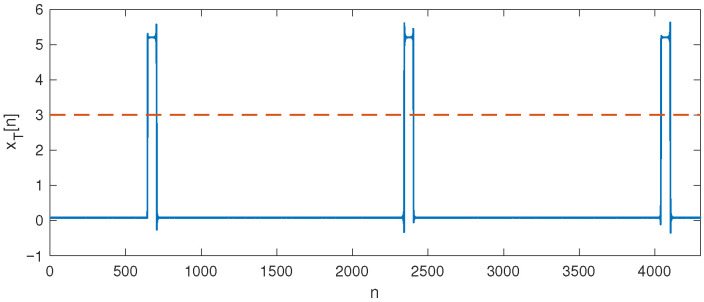
Signal acquired from the output of the optical tacho sensor.

**Figure 2 sensors-25-05669-f002:**
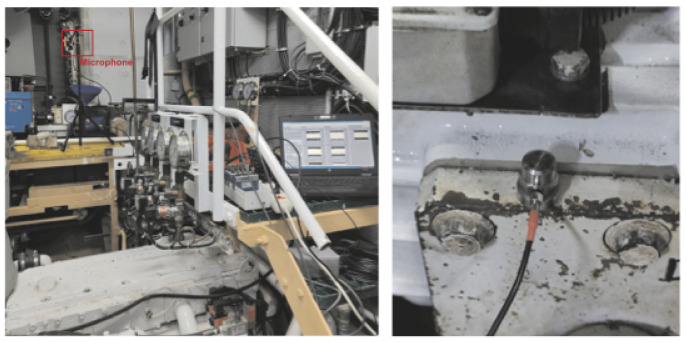
Engine room with measuring equipment and microphone (**left**) and an accelerometer attached to the motor housing with a magnet (**right**).

**Figure 3 sensors-25-05669-f003:**
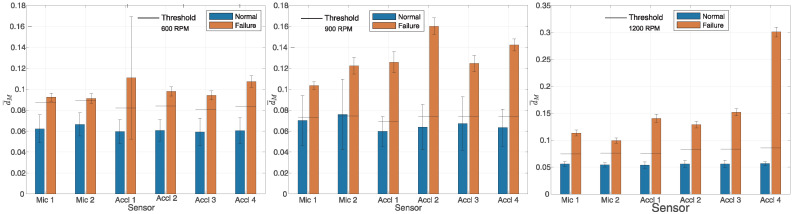
Fault detection at 600, 900, and 1200 RPM, from left to right. First cylinder disabled, N=4096, Hamming window, $N_{T}=100$, and $N_{M}=20$.

**Figure 4 sensors-25-05669-f004:**
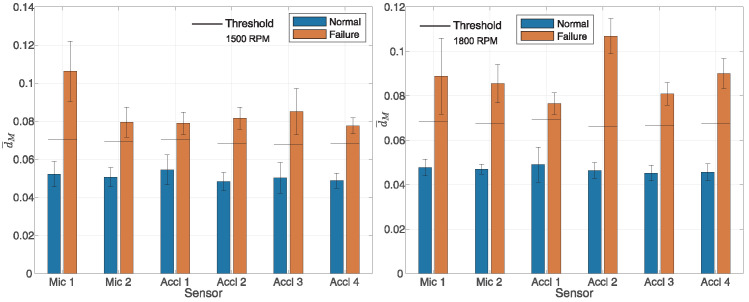
Fault detection at 1500 and 1800 RPM, from left to right. First cylinder disabled, N=4096, Hamming window, $N_{T}=100$, and $N_{M}=20$.

**Figure 5 sensors-25-05669-f005:**
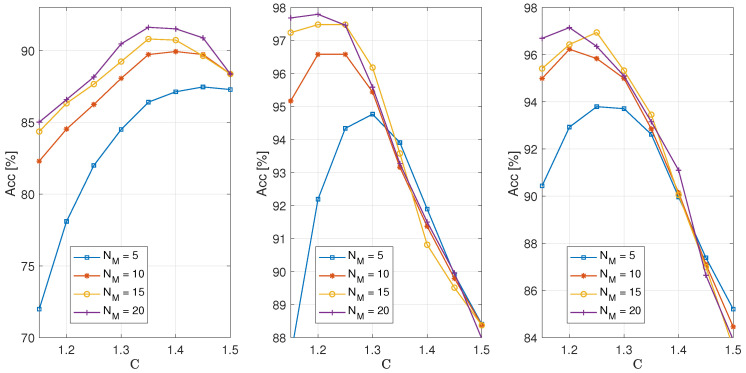
Comparison of accuracy depending on the coefficient *C* for different numbers of full cycles $N_{M}$ used to calculate *d* at $N_{T}$ = 50, 75, and 100 (from left to right) full cycles used for training.

**Figure 6 sensors-25-05669-f006:**
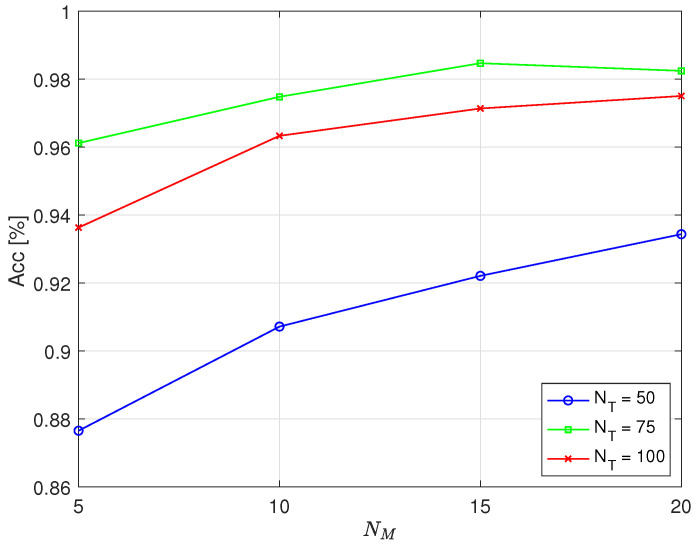
The dependence of the accuracy of the proposed method on the number of full cycles $N_{M}$ used to calculate the parameter *d* for the case of the first cylinder being disabled.

**Figure 7 sensors-25-05669-f007:**
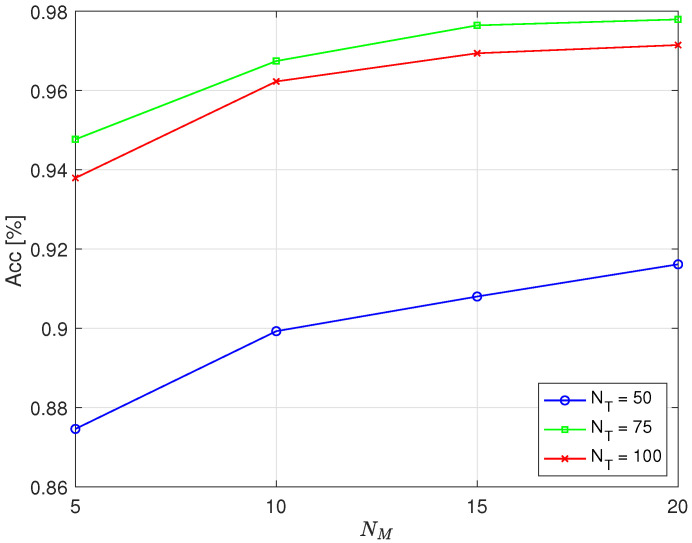
The dependence of the accuracy of the proposed method on the number of full cycles $N_{M}$ used to calculate the parameter *d* for the case of the fifth cylinder being disabled.

**Figure 8 sensors-25-05669-f008:**
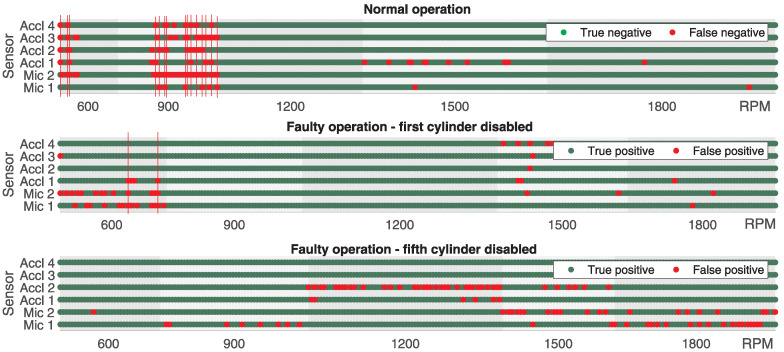
Visualization of classification success for normal and faulty engine operation for all sensors where $N_{T}=50$, $N_{M}=5$, and N=4096.

**Figure 9 sensors-25-05669-f009:**
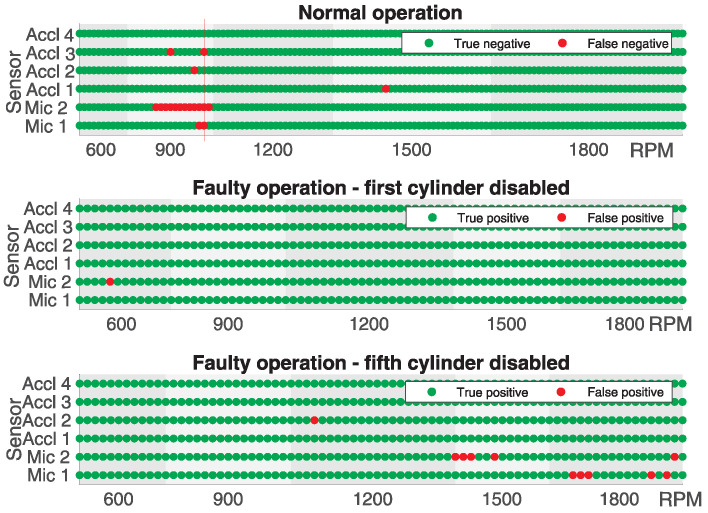
Visualization of classification success for normal and faulty motor operation for all sensors where $N_{T}=75$, $N_{M}=15$, and N=4096.

**Figure 10 sensors-25-05669-f010:**
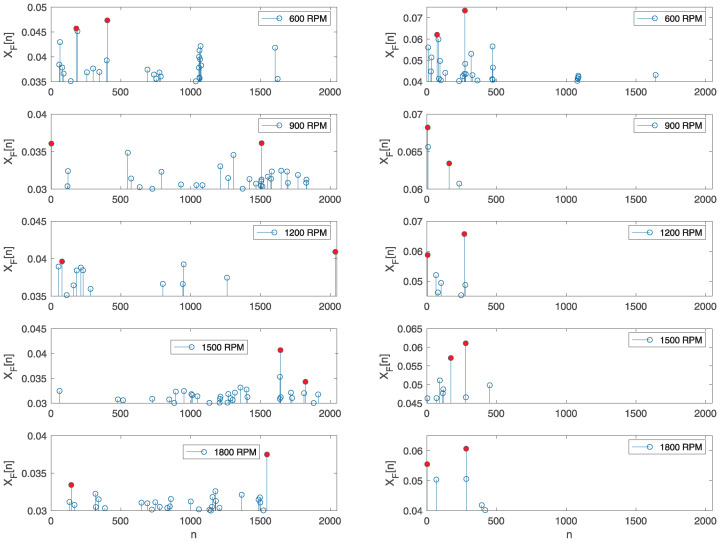
Comparison of the two highest values of the signal samples $X_{F}$, marked with red dots, during normal operation (**left**) and faulty operation (**right**) at different engine speeds, with $N_{T}=75$, $N_{M}=15$, and N=4096.

**Figure 11 sensors-25-05669-f011:**
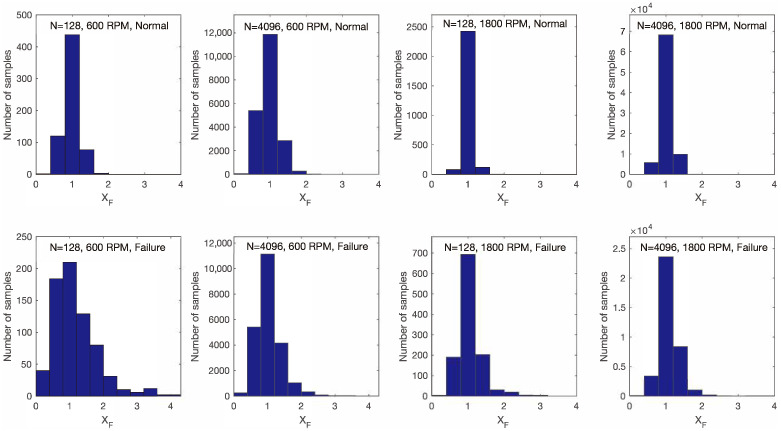
Histograms of the signal samples $X_{F}$ under normal (**top row**) and faulty (**bottom row**) conditions, shown for various rotational speeds and window function size. The bin width was set to 0.4.

**Figure 12 sensors-25-05669-f012:**
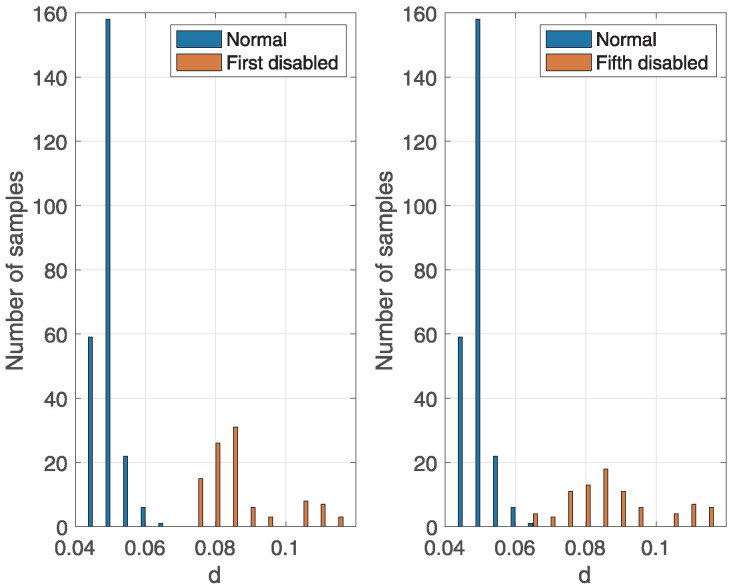
The distribution of the variable *d* for correct and incorrect engine operation when the first or fifth cylinder was deactivated at 1800 RPM, N=4096, $N_{T}=75$, and $N_{M}=15$.

**Table 1 sensors-25-05669-t001:** Overview of the results of all conducted tests.

Cylinder Dis.	$N_{T}$	$N_{M}$	TP	FP	FN	TN	Acc [%]
First	50	5	1244	214	269	2185	87.65
		10	654	72	109	1115	90.72
		15	454	26	74	730	92.21
		20	337	17	46	560	93.44
	75	5	1412	46	100	2204	96.12
		10	724	2	45	1095	97.48
		15	479	1	18	744	98.47
		20	353	1	15	543	98.25
	100	5	1403	55	175	1979	93.63
		10	720	6	60	1014	96.33
		15	467	13	21	687	97.14
		20	353	1	21	507	97.51
Fifth	50	5	1205	217	269	2185	87.46
		10	617	85	109	1115	89.93
		15	425	43	74	730	90.8
		20	314	34	46	560	91.61
	75	5	1327	95	100	2204	94.77
		10	687	15	45	1095	96.74
		15	457	11	18	744	97.64
		20	343	5	15	543	97.79
	100	5	1375	47	175	1979	93.79
		10	695	7	60	1014	96.23
		15	453	15	21	687	96.94
		20	344	4	21	507	97.15

**Table 2 sensors-25-05669-t002:** Precision, recall, and F1-score for the proposed method when $N_{T}=75$ and $N_{M}=15$ full cycles were used for training and detection, respectively.

Faulted Cylinder	Precision [%]	Recall [%]	F1–Score [%]
1st cylinder	99.8	96.4	98.1
5th cylinder	97.6	96.2	97.0

**Table 3 sensors-25-05669-t003:** Dependence of accuracy on the applied window function.

Window Function	Acc [%]
Rectangle	93.64
Chebyshev	96.78
Blackman	98.23
Hanning	98.23
Hann	98.23
Hamming	98.47

**Table 4 sensors-25-05669-t004:** Accuracy dependence on number of samples *N*.

*N*	Acc [%]	
	1. Disabled	5. Disabled
128	95.09	95.69
256	96.7	96.7
512	97.34	97.48
1024	98.23	98.13
2048	98.39	98.21
4096	98.47	97.64

**Table 5 sensors-25-05669-t005:** Accuracy comparison of various methods.

Method	Type of Failure	Accuracy [%]
Feature extraction from wavelets and Bayesian optimization [6]	Engine misfire	100
	Ignition timing variation	85
	Air fuel ratio variation	73
VMD-CWT ^1^ -CNN ^2^ -SVM ^3^ [5]	Insufficient oil supply or	100
	cylinder misfire or	
	six cylinder misfire or	
	clogged air filter or	
	damaged oil supply pipe	
VMD-CWT-CNN-RF ^4^ [5]	Insufficient oil supply or	98.7
	cylinder misfire or	
	six cylinder misfire or	
	clogged air filter or	
	damaged oil supply pipe	
CWT-CNN [12]	Knocking combustion	92.62
WDCNN ^5^ _MMD ^6^ [18]	Misfire diagnosis	97.519
DAWDCNN ^7^ [18]	Misfire diagnosis	99.9
ARVMD-CEDS ^8^ [4]	Valve clearance faults or	98.7
	insufficient fuel supply or	
	abnormal rail pressure conditions	
Proposed method	Disabling 1st or 5th cylinder	98.06

^1^ CWT —Continuous Wavelet Transform. ^2^ CNN—Convolutional Neural Network. ^3^ SVM—Support Vector Machine. ^4^ RF—Random Forest. ^5^ WDCNN—Wide-Kernel Convolutional Neural Network. ^6^ MMD—Maximum Mean Discrepancy. ^7^ DAWDCNN—Domain-Adversarial Wide-Kernel Convolutional Neural Network. ^8^ ARVMD-CEDS—Adaptive Recursive Variational Mode Decomposition and Component Energy Distribution.

## Data Availability

Research data are available at: SharePoint Folder.

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
