# Peer review of "Spectral-Based Fault Detection Method in Marine Diesel Engine Operation"

_sensors, 2025, doi:10.3390/s25185669_

Round 1
Reviewer 1 Report
Comments and Suggestions for Authors
Reviewer’s Report on the manuscript entitled:
Spectral Based Fault Detection Method in Marine Diesel Engine Operation
The authors proposed a method for detecting engine malfunctions through analyzing signals acquired from a microphone and accelerometer. Their method is based on a frequency analysis of the signals using FFT. It defines a measure by which the engine’s operation can be classified as either correct or faulty. They tested their model on a functioning diesel engine where a fault was emulated by deactivating one cylinder. I found the methodology and results interesting; however, the structure, presentation, literature review, and figure quality must be improved. Please see below my comments.
Abstract. Please add some numerical values.
Lines 29-46. The literature review should be improved:
The least-squares wavelet analysis (LSWA) is a time-frequency analysis method that is a natural extension of the least-squares spectral analysis. The LSWA has been used for fault/anomaly detection in interferometry signals and studying the interconnection between cycles of two different signals in the following study, so please review and discuss here:
Least-squares Wavelet and Cross-wavelet Analyses of VLBI Baseline Length and Temperature Time Series: Fortaleza–Hartebeesthoek–Westford–Wettzell (Publications of the Astronomical Society of the Pacific, 2021)
The following recent study also proposed a compressed sensing model for efficiently capturing the essential features of vibration signals under rotational speed variations, so please review and discuss:
Compressed Sensing of Vibration Signal for Fault Diagnosis of Bearings, Gears, and Propellers Under Speed Variation Conditions (Sensors, 2025)
Other models that have advantages over FFT and are used for seismic signals, hydrophone signal processing include Multichannel Antileakage Least-Squares Spectral Analysis (MALLSSA), Antileakage Fourier Transform (ALFT), etc. These spectral models can also be reviewed.
Lines 74-78. Please first mention the research gaps, then highlight the main contributions of your research, preferably using a few bullet points.
I suggest using the following structure for better readability and flow:
1.Introduction, 2. Materials and Methods, 2.1. Datasets, 2.2 Methods, 3. Results, 4. Discussion, 5. Conclusions. Please then update lines 89-91.
Line 92. Please move the description of the dataset from your current Section 3 to the beginning of Section 2 in a subsection. Please follow the structure that I suggested above.
Lines 121 and 122. Please provide a rationale on how the resampling is done here. What is the type of resampling? Is it linear, quadratic, spline, etc.?
Line 185. Please change the title of this section to “Results”. Then please add a Discussion section before the conclusion section. In the discussion section, please discuss/compare your results with literature including the articles that I suggested above (LSWA, LSSA, MALLSSA, ALFT, etc.). Please also elaborate on uncertainties involved in the input data, modeling and parameters, etc. and mention the limitations of your research and provide future direction/recommendations.
Line 251. Please add a reference for these metrics. Also, why not calculating, precision, recall, and F1-score in addition to accuracy? Please add these in a table.
Algorithm 1. What are the stoppage criteria for this loop?
Line 404. Since you are adding a separate discussion section, please rename this section only “Conclusions”.
Line 418. Where you wrote: “From the obtained results, it can be concluded that the method is universal, whether analyzing acoustic or vibration signals…”. This is a strong statement. The experimental results should be improved. I suggest artificially introduce noise and disturbances into the signals and estimate their spectra and elaborate further on the results.
Other comments.
Line 91. Please capitalize “F” in “Finally”.
Figures 3 and 4 have tiny font size. Figure quality must be improved. Please enlarge the font size and ensure the figures have a resolution of at least 300 dpi.
Please enlarge the font size of Figures 5,6,7, etc.
Please DO NOT use italic font for variables/subscripts/functions that have more than one letter. Please check algorithms and equations.
Thank you and regards,
Comments on the Quality of English LanguagePlease carefully review and correct typo/style/grammar issues.
Reviewer 2 Report
Comments and Suggestions for Authors
This paper proposes a frequency domain signal analysis method based on the use of DFT for the fault diagnosis of diesel engines, and verifies the effectiveness of the method through experiments. There exists some worries to be solved before it suitable for publication.
- The author claims that the accuracyof fault diagnosis using acoustic signals and vibration signals respectively are similar. The author needs to explain the necessity of combining acoustic signals and vibration signals for fault diagnosis.
- The references cited by the author are rather outdated. Please appropriately quote the latest literature from the past three years.
- In Figure 6 and Figure 7, considering that the optimal value of parameter C is related to NT, it should be further explained why C=1.2 is chosen here?
- The contributionsof this paper should be clearly pointed out in Introduction
- There are some writing mistakes here: In line 91, the letter "f" should be capitalized. In line 136, should the symbol |Xw[0,m]| be written as |Xw[k,m]|?
- The proposed methodin this paper should be compared with other advanced technologies to ensure its superiority.
Round 2
Reviewer 1 Report
Comments and Suggestions for Authors
Dear authors,
Thank you for addressing my comments satisfactorily and improving your manuscript. I have a few minor editorial suggestions that can also be made during the proofreading if approved by the editor:
There are some references listed in the reference list but not referred to in the texts. For example:
Lines 499-508. Please make references to [21], [22], [23] for LSWA, ALFT, MALLSSA, etc. and other references as applicable.
Some of the references have doi and some do not. Please add doi/url to all the references.
Lines 102, 115, 119, 123 the headings in bold should have spaces between the words, for example "Unifiedprocessingofacousticandvibrationsignals" should be "Unified processing of acoustic and vibration signals".
Please carefully proofread the manuscript and correct style/grammar/typo issues.
Thank you and regards,
Reviewer 2 Report
Comments and Suggestions for Authors
There exists some worries before it suitable for publication:
1.The article should be compared with other advanced algorithms, and the conclusion of the article should be supported by experimental data.
2.In line 111, should the word "Novelfrequency-domainfaultmeasure" be corrected to "Novel frequency-domain fault measure" ? The question that has been clarified also appears in lines 102,115,119,123.
